# Clinical Indications and Compassionate Use of Phage Therapy: Personal Experience and Literature Review with a Focus on Osteoarticular Infections

**DOI:** 10.3390/v11010018

**Published:** 2018-12-28

**Authors:** Olivier Patey, Shawna McCallin, Hubert Mazure, Max Liddle, Anthony Smithyman, Alain Dublanchet

**Affiliations:** 1Service of Infectious and Tropical Diseases, CHI Lucie et Raymond Aubrac, 94190 Villeneuve Saint Georges, France; opatey@aol.com; 2Department of Musculoskeletal Medicine DAL, Centre Hospitalier Universitaire Vaudois CHUV, Service of Plastic, Reconstructive & Hand Surgery, Regenerative Therapy Unit (UTR), CHUV-EPCR/Croisettes 22, 1066 Epalinges, Switzerland; shawna.mccallin@gmail.com; 3HGM Consultants, 63 Rebecca Parade, Winston Hills, NSW 2153, Australia; humazure@hotmail.com; 4School of Life Sciences, University of Technology, Ultimo, NSW 2007, Australia; max.liddle17@gmail.com; 5Cellabs Pty Ltd, and Founder Special Phage Services Pty Ltd, both of 7/27 Dale St, Brookvale, NSW 2100, Australia; tonysmithyman@gmail.com

**Keywords:** bacterial infection, antibiotic resistance, bacteriophage, antibiotic therapy, phage therapy, cases report

## Abstract

The history of phage therapy started with its first clinical application in 1919 and continues its development to this day. Phages continue to lack any market approval in Western medicine as a recognized drug, but are increasingly used as an experimental therapy for the compassionate treatment of patients experiencing antibiotic failure. The few formal experimental phage clinical trials that have been completed to date have produced inconclusive results on the efficacy of phage therapy, which contradicts the many successful treatment outcomes observed in historical accounts and recent individual case reports. It would therefore be wise to identify why such a discordance exists between trials and compassionate use in order to better develop future phage treatment and clinical applications. The multitude of observations reported over the years in the literature constitutes an invaluable experience, and we add to this by presenting a number of cases of patients treated compassionately with phages throughout the past decade with a focus on osteoarticular infections. Additionally, an abundance of scientific literature into phage-related areas is transforming our knowledge base, creating a greater understanding that should be applied for future clinical applications. Due to the increasing number of treatment failures anticipatedfrom the perspective of a possible post-antibiotic era, we believe that the introduction of bacteriophages into the therapeutic arsenal seems a scientifically sound and eminently practicable consideration today as a substitute or adjuvant to antibiotic therapy.

## 1. Introduction

In 1917, Félix d’Hérelle observed a phenomenon in stool cultures from convalescent patients with bacillary dysentery [1], which took the form of perfectly round clear areas in the bacterial lawn. He made the assumption that these clear zones were caused by an "invisible microbe" capable of killing bacteria, to which he gave the name bacteriophage. Two years later (1919), he demonstrated that the oral administration of bacteriophages in humans is harmless and causes the healing of bacterial enteritis caused by *Shigella* sp. (bacillary dysentery). Based on a large number of published cases in the years that followed, the interest and use of this new treatment in various infections spread rapidly across the world, reaching nearly every continent [2,3]. This was the situation until the discovery of antibiotics; when faced with their easier use, phage therapy was gradually abandoned in Western countries until it finally disappeared completely in France with the closure in 1990 of the elast remaining sources of therapeutic bacteriophages from the two Pasteur Institutes (Paris and Lyon). However, phage therapy continued uninterrupted in the Soviet Union during this time and is still practiced in Russia, Poland, Georgia and some other former Soviet States today, in accordance with specific national regulations.

We have been witnessing the worldwide spread of multidrug-resistant (MDR) bacteria in recent years. As new and truly innovative antibiotics are rare, the increasing frequency of therapeutic failures are raising fears of a new pre-antibiotic era [4]. To respond to this worrying situation, the return of phage therapy seems to be an answer not only as an alternative [5], but also a complementary treatment, to faltering antibiotic therapy [6,7,8]. This renewed interest in phage therapy is manifested by the motivation to conduct several clinical trials since 2009 that have used phages for a variety of indications, including chronic otitis, burn wound or urinary tract infections (UTI), and *Escherichia coli* diarrhea [9,10,11,12]. Indeed, phage therapy must be proven to be therapeutically effective through experimental clinical trials in order to obtain marketing approval, which is required for use in Western medicine. While studies have repeatedly documented its safety, it is unfortunate that no marketing approval has been attributed to a phage product to date as a result of these resource-intensive studies; three trials were unable to statistically prove efficacy [9,11,13], even if clinical benefit was achieved for some patients, and the only trial that was successful has not been further pursued for commercialization [12].

Many researchers and medical doctors have voiced the need to revise the regulatory classification of phage therapy products in order to facilitate their clinical evaluation. Natural phages are currently classified as Medicinal Products (MP) under European Union (EU) legislation [14] and as a drug by the Food and Drug Administration (FDA) in the United States, which necessitate that phages be produced under Good Manufacturing Practice (GMP) guidelines and infrastructure. While such criteria do not completely inhibit the ability to conduct trials, they do render formal phage trials more difficult and more expensive to conduct. Substantial financial investment is required to conduct clinical trials, and an inconvenient amount of time is needed to procure results in order to address current clinical needs.

Phage therapy is now at a state where it is not officially recognized as a legitimate treatment, but has been increasingly granted emergency approvals for addressing antibiotic treatment failures. There are more than 10 published case reports [15] and a fast-growing number of undocumented compassionate cases that report successful treatment outcomes with phage therapy. Within only the last year, two experimental phage therapy centers have opened in addition to the long-established Phage Therapy Unit at the Ludwik Hirszfeld Institute in Poland (Box 1). From this perspective, and in conjunction with a century of publications on this subject for different bacterial infections treated by phage therapy, compassionate use and case reports constitute an invaluable source of knowledge that help to elucidate best practices for phage therapy. Many original and historical texts published in French or Russian have been unfortunately excluded from contributing to this large body of information and should enter into consideration. While case reports and historical accounts do not substitute for formal clinical trials, the findings and remarks they contain are useful to set up modern therapeutic protocols and hopefully to avoid conducting additional therapeutically-futile clinical trials. This is what we propose to report here, both by summarizing findings from the literature and by adding our own experience of cases, particularly for osteoarticular infections, treated under compassionate protocols, as a means to advise on the progression of phage therapy into modern medicine.

Box 1Experimental phage therapy centers established in Western countries.Ludwik Hirszfeld Institute of Immunology and Experimental Therapy, Polish Academy of Sciences, Phage Therapy Unit in Wroclaw, Poland (IIET PAS PTU): This is the oldest and most established experimental center in central Europe, which has been preparing phage formulations for hospital use in Poland since the 1970s, before it became a member state of the European Union (EU). Phage therapy was and is continued under the national regulatory framework as an experimental therapy for specific medical conditions and in specific centers under Article 37 of the Helsinki Declaration. They have an in-house bank of phages against 15 different bacterial pathogens (*Staphylococcus, Enterococcus, Pseudomonas, Escherichia, Klebsiella, Serratia, Proteus, Acinetobacter, Citrobacter, Enterobacter, Stenotrophomonas, Shigella, Salmonella, Burkholderia, Morganella*). Treatment is proceeded by phage susceptibility testing (phage typing procedure) and preparations are used for outpatient treatment. The PTU periodically publishes summaries of their experiences [16,17,18,19,20] that provide factual justification for using phage therapy and useful information for clinical applications.Magistral preparations in Belgium (also known as a compounded prescription drug in the US) [14]. Phage therapy can be provided as a magistral preparation in Belgium since 2018, after several years of discussion involving public health and federal regulatory authorities, in order to facilitate physician-prescribed treatment for individual patients. Phages are considered active pharmaceutical ingredients (APIs) that must be produced according to an internal monograph (set of instructions) and that are subsequently certified by competent laboratories before they are mixed or put into formulation under the supervision of a pharmacist and delivered to a specific patient.Center for Innovative Phage Applications and Therapeutics (IPATH) University of California San Diego, School of Medicine: This center announced its opening in June 2018 following several successful treatments with phage. Their first case used phage to treat an MDR systemic infection caused by *Acinetobacter baumannii*, which was initiated and coordinated by the wife of the patient, a global health professor, and his physician [21,22]. While this was the first American patient with a systemic MDR infection to be successfully treated intravenous (iv) by phage therapy, more than five patients have been treated since under the FDAs compassionate use program and IPATH is planning to conduct clinical trials in the near future.

## 2. General Prerequisites for the Medical Use of Bacteriophages

Due to the unfamiliarity with and particularities of phage therapy, it is worthwhile to touch upon several general aspects of clinical use: product availability, production, formulation and administration, dosage, and evaluation. The permission to use phage therapy for compassionate or experimental treatment, at the patient, physician, hospital, and health and regulatory authority levels, are beyond the scope of this publication, but are evidently necessary to proceed with treatment and requirements may vary country-to-country. Approvals are now often granted on an individual bases for emergency use or in the case of antibiotic treatment failures, mostly in France, Belgium, Poland, Australia, and the US.

### 2.1. Availability

The first condition for use of phage therapy is simply to have bacteriophages available for treatment, which is often complicated at this stage of phage development. This implies having access to phages that are both biologically active against the patient’s bacterial isolate and satisfy regulatory requirements (purity, traceability, characterization). A single phage may be used (monophage preparation) or several phages may be combined against one or more bacterial species (phage cocktail).

As phage therapy is not currently a recognized medicine in the West and no registered products exist, phages are either being prepared specifically for a patient infection (personalized or custom approach) or treatment can be done with commercial phage preparations from Russian or Georgian suppliers, which have pre-defined phage compositions (“ready-to-use”) [23]. Such commercial bacteriophage preparations that are available for purchase may, or may not, encounter importation difficulties into Western countries due to product traceability or a lack of certification or analytical information. Access can be accomplished by patients traveling to countries where phage therapy is an approved practice (medical tourism), and the Eliava Institute in Georgia treats a number of foreign patients onsite each year. This last option, however, is dependent upon patient mobility and financial ability to pay for treatment.

Alternatively, phages have been prepared for compassionate cases by small biotech and academic institutions for individual patients. Indeed, a large number of different bacteriophages are deposited in different collections that target clinically relevant bacteria, and it would be desirable that the collections held in these "phage banks" organize themselves into a network to facilitate exchanges. A networking initiative, known as the Phage Directory [24], is attempting to facilitate phage sharing for emergency or compassionate clinical needs. If an active phage is not present or available from such an organization, it is normally still possible to isolate one from the environment, although this is pathogen dependent, sufficient characterization is still required, and is difficult to achieve for acute life-threatening infections [25].

### 2.2. Production

Phage products must be produced with an acceptable level of purification for clinical use in order to remove remaining endotoxin and bacterial contaminants. If phage preparations are viewed as medicinal products, they will be subject to GMP compliance, which are standards intended to guarantee the quality of a medicine [26]. This requires that a procedure be defined for their manufacture and stipulates a combination of physicochemical and biological tests, as well as stringent production facilities. The quality (i.e., the stability and consistency) of a biological drug, such as phages, is harder to guarantee and control than that of a chemical, and GMP requirements have put a strain on the clinical development of phage therapy in Western medicine, as well as greatly increasing production costs. Indeed, GMP constraints both delayed patient enrollment for the Phagoburn clinical study and negatively impacted the phage titer of the final product [9]. An adaptation of the regulation is necessary [27,28,29,30,31,32] and, in particular, will have to take into account the use of individualized preparations [33] as a personalized medicine [28,34], and modification of phage components throughout treatment to counteract bacterial–phage resistance. Production considerations must take into account the sustainability of the phage treatment approach and patient safety.

Phage therapy is currently implemented for compassionate use and individual patients by by-passing GMP-requirements. Belgium has opted to facilitate phage therapy by presenting the phage as magistral preparations, which are individually prepared by prescription for individual patients by a qualified hospital pharmacist, and the quality of phage preparations are verified by accredited laboratories (Box 1) [14]. Even without such a systematic approval system, phage biotech companies (MicroGen, Eliava, Pherecydes Pharma, Advanced Phage Therapeutics, AmpliPhi Biosciences), as well as academic institutions and military research institutions, have helped in the production process and/or supply of phages for emergency use.

### 2.3. Formulation and Administration

The administration of a drug is dependent on the vectorization/formulation of the active phage component [35,36]. Local application is easiest to apply, and tolerance has been repeatedly documented for this route [9,37]. Phages may be applied topically either in cream/gel formulations or by contact with soaked bandages on the wound surface. Bacteriophages, being of a protein nature, raise the concern of an anaphylactic reaction following repeated administration. However, severe reactions have only been rarely reported, and today the risk is further reduced by advanced purification methods [38,39,40].

While oral administration is easy and without side effects, gastric acidity is a hostile barrier to ingested bacteriophages. To overcome this drawbackthere are two possible approaches: alkalinisation, by administration of an alkaline liquid (bicarbonate water, carbonated water), or gastro-resistant vectorization (i.e., in release capsules or pills). Alkaline neutralization may come with an increased risk of opportunistic infections for patients, and vectorization comes at an increased cost of production; more clinical data are required to determine the best strategy for oral phage application.

Inhalation seems to be effective in delivering lyophilized bacteriophages to the lungs in the form of powder propelled by inhalers [41]. The bronchopulmonary tree is indeed easily accessible by air, and thus, it is conceivable to spray bacteriophage suspensions (nebulization, misting) or dry forms (spray) [42]. However, few cases have been published to date [43,44,45,46,47,48], and using appropriate vectorization for inhalation remains to be evaluated.

Diffusion is rapid after systemic ie intravenous administration, though circulating phages are sequestered by the reticuloendothelial system in the spleen and liver. In the absence of bacterial target hosts, phages are quickly eliminated. On the contrary, if bacterial targets are present, phages multiply to a degree dependent on a multitude of bacterial (metabolic activity, sensitivity) and mammalian factors, making the estimation of pharmacokinetics [49] variable and difficult to estimate between patients.

### 2.4. Dosage

The required dosage, rhythm, and duration of treatment have been poorly studied. Theoretically, in situ multiplication requires only one application; while in practice, repetition is often the rule. Unlike conventional drug treatments, the pharmacological parameters are poorly defined and understood at present, which presents the main difficulty in being unable to predict the extent of in vivo multiplication.

### 2.5. Therapeutic Evaluation

Like any drug, a biomedicine must be studied experimentally to appreciate its positive and negative effects on a living organism. Although many publications (individual cases and clinical series) have shown positive results of phage therapy and presented few adverse effects, it is necessary to respond to modern requirements and to carry out randomized, double-blind controlled trials [50,51]. Nonetheless, simpler hospital observational studies, despite their drawbacks and inadequacies, would make it possible to provide highly valuable information for pressing questions while satisfying prerequisites (i.e., the number of patients likely to be included within a defined period of time) that have often been difficult for modern trials to achieve to date.

Many case studies today evaluate phage therapy by the most essential factor: the clinical improvement of the patient. However, information documenting phage activity within the patient, such as phage amplification or phage sensitivity, are often lacking, and therefore claims that phage therapy causes clinical amelioration are not data-supported. Much more information could and should be obtained from compassionate and emergency-use treatments to further our knowledge-base of phage therapy in humans.

## 3. Clinical Indications in the Literature 

Inherently, phages are able to treat any clinical presentation of bacterial infections. Reports have been published using phage therapy for a large array of clinical indications, including gastro-intestinal [2,11,52,53,54], localized [3,37,55,56,57,58,59], burn wound [9,60], systemic [21,39,61,62,63,64,65,66,67,68,69,70,71,72], urogenital [10,73,74,75,76,77,78,79,80,81,82], respiratory [44,45,47,82,83,84,85], oto-rhino-laryngeal (ORL) [12,86,87,88,89,90,91,92,93,94,95], and osteoarticular infections (see section below). These clinical indications include infections that are acute or chronic, sensitive or resistant to antibiotics, and are caused by highly variable common or opportunistic pathogens.

Acute systemic infections, such for septicemia or meningitis, have been treated with phages with some success. It seems premature to consider such indications initially for phage therapy for at least two reasons: the urgency of treatment and the need for parenteral administration. Both aspects require readily-available, highly-purified phages, and rapid approval processes that, while not insurmountable, are not feasible for broad implementation at this time. Indeed, the few published cases of systemic treatments are the result of a few geographical competency centers and close collaboration between phage researchers and clinicians.

Chronic infections, however, are increasingly frequent and have gained attention as a target for phage therapy. Chronicity is supported by the formation of bacterial biofilms, intracellular bacterial persisters, or tolerant bacteria that are particularly problematic for UTIs, bacterial prostatitis, prosthetic joint infections (PJI), osteomyelitis, and respiratory conditions such as cystic fibrosis (CF). They require long-term antibiotic treatment that disrupts healthy microbiomes and selects for antimicrobial resistance. Such infections, if not constantly suppressed, risk development into bacterial sepsis. CF, although not an infectious disease itself, is the subject of special attention for phage therapy because of the chronic state of repetitive superinfections in these patients, which are usually caused by mucosal *Pseudomonas aeruginosa* strains resistant to many antimicrobials and capable of forming biofilms [96].

In addition to classical pathogens, opportunistic bacteria are often multidrug-resistant and cause infections that are difficult to control [96], for which the question of the interest of phage therapy is repeatedly raised. Infections with some bacteria, such as mycobacteria, present additional biological obstacles, such as preferential intracellular location of bacteria (macrophage or epithelial cells) and a slow growth rate, as in tuberculosis. Ready-to-use bacteriophage suspensions are generally not available for such situations, and a few teams have looked at some of them, although it is still too early to draw any conclusions. These include, more specifically, infections caused by *Helicobacter* [97,98]; *Borrelia* (Lyme disease [99]), as well as *Brucella*, *Yersinia pestis* and *Bacillus anthracis* [100] in the context of biological-weapon risks [101]. Other bacterial species (*Haemophilus influenzae*, *Streptococcus pneumoniae*) do not generate much interest today for phage therapy application. It should be noted that *Campylobacter* bacteriophages are mainly studied in poultry farms in a preventive context rather than therapeutically.

Osteoarticular infections are a particular form of deep-seated, localized infection that are a prime target for phage therapy given their frequency and poor response to antibiotic therapy. The diffusion of antibiotics into bone tissue is often mediocre and impaired by the presence of bacterial biofilms that form *in vivo* at the contact between bone and prosthetic material. The recurrence and transition to chronicity is more and more common for many reasons, including the presence of MDR bacteria. Today, the number of post-surgical bone infections on fracture or joint prosthesis continues to increase [102]. Conventional antibiotic treatments are long and costly, with frequent repeat surgery, and sometimes amputation is the only infection control option [103].

Phage therapy has been used very early and frequently for this type of infection, as evidenced by many publications from Northern America [104] and in Eastern European countries [105,106,107]. In France, the surgeon André Raiga [66,108] made several assessments of his long experience in this field. Clinical cases in Strasbourg, France were published in 1979, which document positive outcomes in bone infections with phages (Box 2) [109]. A review of Soviet literature has also indicated complete recovery from osteomyelitis using phages alone or in combination with antibiotics [110]. More recently, two cases of PJI (*Staphylococcus aureus*) and one case of *P. aeruginosa* osteomyelitis were treated with direct application of phages in France [111].

The potential of phage therapy to treat such post-accidental, surgical osteitis, or peri-prosthetic joint infections is likely rooted in phage activity against bacterial biofilms and potentially against intracellular bacteria. An experimental model [112] has demonstrated that a treatment combining bacteriophages and antibiotics helps to dissolve biofilms with a pronounced effect on biofilms of *Staphylococcus* sp. compared to those of *P. aeruginosa*. Indeed, there has been a very large number of experimental studies for several years on this subject not only in vitro, but also in vivo [113,114]. While bacterial infections begin by biofilm formation on prosthetic surfaces, they can become chronic by establishing an intracellular life-style within mammalian cells that shields them from antibiotic treatment and then causes recurrent active infections. A recent model documented the ability of phages to kill intracellular *S. aureus* [115].

All of the above provide substantial evidence that osteoarticular infections are a sound target for phage therapy. With this logic, a budget has been attributed for a future clinical trial in France, “Phagos,” for PJIs caused by *S. aureus*, which will begin as soon as GMP-compliant phage suspensions are achieved [116]. Our experience with the compassionate treatment of oesteoarticular, as well as other, infections is presented herein.

Box 2Conclusion of Lang et al. 1979 [109].Seven orthopedic surgery cases were treated with bacteriophages between 1975 and 1976. Of the treated patients, six were male and one was female. The age of patients ranged from 19–70 years of age. The cases presented by authors were chronic, having exhausted the usual therapeutic arsenal, and phage was added to other treatments in order to maximize patient benefit.Five treatments resulted in good clinical outcomes, which was supported by radiological and bacteriological examination. A condition was considered improved if symptoms were ameliorated and radiological examination was positive, but problems persisted with scarring and positive bacterial cultures (one case). Treatment failure with added phages occurred for one patient and caused a change in treatment plan, comprising first local and general antibiotic therapy (ampicillin, cephalosporin, gentamicin), then hyperbaric oxygen therapy, and finally surgical intervention, which ultimately resulted in a favorable outcome. In conclusion, the use of suitable bacteriophages in the treatment of antibiotic-resistant chronic bone infections seemed to be an interesting therapeutic alternative for authors, and the results of these cases encouraged continuation in this therapeutic direction.

## 4. Compassionate Phage Use in France and at Villeneuve Saint Georges

Phage therapy was used to treat patients compassionately during the 1970s and 80s in France, at a time when it was possible obtain suspensions of therapeutic phage for the pathogenic bacterium of a patient from the Pasteur Institute. The clinical outcomes during this time with the treatment of frequent, high-risk infections with phages have been summarized previously in a short paper outlining conclusions and new indications for phage therapy (Box 3) [117]. At that time, phage therapy was routinely performed in some hospitals, such as in Lyon, Paris, and Strasbourg. A surgical service at the latter had published a small clinical study of seven cases and concluded that phage therapy was promising, particularly in bone infections (Box 2) [109]. Several patients with bone infections in the hospital of Villeneuve Saint Georges, for whom conventional treatment had failed, also benefited from such phage therapy treatment during this time (unpublished results). However, by 1990, phage therapy and its practice in France became impossible after phage production was ceased at the Pasteur Institute. There followed a period of about 15 years during which phage therapy was totally inaccessible in France.

In 2004, we were able to buy over-the-counter commercial preparations of bacteriophages from pharmacies in Moscow for a few dozen Euros. After an evaluation (for sterility, activity, specificity) of these preparations [118], those capable of responding to the clinical problems at hand were retained and used. The first case we treated was a particularly worrying case of an evolving infection of the external auditory canal, where a bacteriophage suspension against *S. aureus* was used to treat chronic otitis externa (Box 4; Patient 1 in Table 1). With this experience, and in the face of the increasing therapeutic failures that we were confronted with, especially in orthopedic surgery, some of us decided to reintroduce phage therapy more routinely from 2008 in the hospital in which we practiced, and it is still occasionally used as needed at the hospital of Villeneuve Saint Georges. We will briefly outline the process of using phages for compassionate use and present several clinicals cases of our experiences in phage therapy.

## 5. Protocol for Compassionate Use of Phage Therapy

Before patient admission, the decision to use phage therapy is made by a multi-disciplinary hospital team (surgeon, infectious disease specialist, microbiologist), who conduct a complete examination of the patient and patient file. In addition to the biological assessment, one or more preliminary specimens is taken to isolate the bacterium and test its sensitivity to available phages. Patients are informed about phages and the possibility of treatment. Phages are administered by a treating physician who exercises their ethical right to use an experimental treatment in the best interest of the patient, without an elaborate regulatory or administrative framework.

During therapeutic care, if necessary, the infectious foci are excised (debridement) and cleaned in the operating room. One or more intraoperative specimens are collected to confirm the initial bacteriological diagnosis. At the end of surgery and before closure of the operative field, the preparation of bacteriophages is used to flood the operative field (5 to 10 ml according to the surface of the field). Access to the treatment site (opening or drain) allows a bacteriological control and the introduction of the same phage preparation in the days following the intervention.

Antibiotic therapy reflecting the pathogen’s antibiotic resistance profile is used in combination with phage therapy, and the patient is kept under surveillance for several days (less than one week) to ensure that there was no evidence of infection (local, biological, or bacteriological). The postoperative course has presented no complications, and no side effects have been reported.

Regarding follow-up, ambulatory monitoring is performed in our facility for several months at a variable frequency, as deemed necessary. The evaluation of each case is performed clinically, as well as biologically and radiologically. Some patients provide us periodically with their health status, which so far has been excellent.

All cases reported here (Table 1) have benefited from compassionate phage therapy for the duly recorded treatment failure. The phage therapy treatments were carried out between 2006 and 2018 after a long evolution, generally several years, of a conventional treatment according to official medical guidelines. All patients had benefited from multiple attempts at treatment (surgical interventions and antibiotic therapy) and had been in therapeutic failure for months or even years. Some had previously tried treatment at the Eliava Institute in Tbilisi. All presented cases were treated in France at the Villeneuve Saint Georges Hospital, unless otherwise noted. The authors have also been involved in the treatment with phage therapy for a case for a refractory UTI in Australia, published previously [78].

The infectious sites were predominantly osteoarticular (9/15), but also included two cases that involved the prostate and other four various infections (two ENT, one abdominal, and one GI tract). The predominantly targeted bacterial species was *S. aureus* (12/15). More rarely, *P. aeruginosa* (three instances) and two instances of *E. coli* were the causative pathogens or were present in polymicrobial infections. Most often, this was a mono-microbial infection (13/15). Suspensions of bacteriophages were mainly from commercial sources (Microgen in Russia and the Eliava Institute in Georgia). In the absence of commercially available preparations, two cases were treated with personalized bacteriophage suspensions.

This small series of cases calls for some remarks. We found that the local application of bacteriophages is completely safe, and no accidents or incidents have been reported. We have also observed highly satisfactory results, and often with rapid improvement. In fact, 12/15 cases resulted in a complete recovery (a secondary problematic pathogen emerged in one case, only a stable condition was achieved for one patient, and one case of a GI infection improved after phage treatment, but for which the condition was not fully resolved). The administration of bacteriophages had always been accompanied by antibiotic therapy with the aim of obtaining a possible synergy. Note that these were chronic cases which had exhausted the usual therapeutic resources, and whose clinical condition was worrying with a poorly functional prognosis. The focus was not to try to experiment or optimize phage therapy, but instead to treat patients with all available resources.

The pathologies that have been treated are varied. In our small case study, bone infections were the most frequent and generally evolved favorably within a few weeks. If there were fistulas, they disappeared, and bone consolidation was observed both clinically and functionally and was confirmed by imaging. Bacterial pathogens became quickly undetectable by microbiology after phage therapy began. After a follow-up for some patients of over 10 years, no relapse has been observed, and it is possible to conclude that patients were completely healed. In two cases where amputation of the lower limb was being considered, this option was avoided. Treatments that prevented the ablation of prosthetic material were also clinically satisfying.

To emphasize the treatment of two prostatitis cases, which constitute the most recent that we have taken care of, infections were caused by *E. coli* in one case and *P. aeruginosa* in the other. They affected elderly people who had been undergoing antibiotic therapy for several months. Concomitant oral and rectal administration over two consecutive days in one case and over three days for the other quickly resolved the recurrent infectious problem.

Box 3Conclusion of Vieu et al. 1979. [117].This article, published in French, highlighted how and why phage therapy was used in France at this time. In particular, the growing importance of opportunistic bacteria resistant to antibiotics in infectious pathology oriented the therapeutic applications of bacteriophages to three new areas: (1) the curative treatment of postoperative surgical infections; (2) suppression of the infectious process during gram-negative pediatric epidemics, caused notably by *Salmonella*, *Klebsiella*, *E. coli*, and *Serratia,* via oral phage administration; and (3) curative treatment of chronic UTIs. The authors noted that a close collaboration between phage scientists and clinicians was absolutely necessary to treat patients with phages, from identifying phages to following clinical progression over time. The success of phage therapy was dependent upon verifying in vitro susceptibility prior to treatment. If treatment failure occurred, it was attributable to low titers of the phage, pH environment of the GI or urinary tracts, inactivation of the phage by simultaneously-prescribed local antiseptics, or the involvement of several pathogens not identified at diagnosis outside the bacterial host range of the phage preparation. In conclusion, the authors affirmed that phage therapy was a merited treatment option due to the frequent clinical successes it produced.

Box 4Treatment of an external otitis.A young patient was examined for chronic otitis after episodes of repeated otitis treated with various antibiotics. The specialist noted an otorrhea and decided to treat it medically (cefpodoxime and ofloxacin) before surgery. Repair of the tympanic membrane was performed. The immediate treatment outcome was obvious: symptoms (pain, drainage) rapidly disappeared with no complications or side effects.After three months, the otorrhea reappeared. The examination was particularly difficult because of very sharp local pain, as the eardrum was inflamed and wet. The resumption of local antibiotic therapy (bacitracin) helped reduce pain. During one year, the patient experienced several treated otorrheas (ofloxacin). During an outpatient consultation with acute pain and under general anaesthesia, a specimen was collected showing the presence in pure culture of *S. aureus* (penicillin-R, methicillin-R, erythromycin-R and ofloxacin-R). Despite antibiotic therapy being immediately prescribed (not specified), the purulent flow and pain persisted, and the *Staphylococcus* was still present.It was then decided to carry out a more precise examination and to collect multiple specimens (tympanic membrane, cutaneous coating of the external duct) before the local application of a bacteriophage suspension, active in vitro against the patient isolate, in combination with pristinamycin. Within 48 hours, the patient noticed a clear improvement: the cessation of purulent flow and pain. Subsequent consultations confirmed a favorable course: the absence of otorrhea or pain and disappearance of *Staphylococcus*. After three months, the ear examination was still very satisfactory and the treatment was stopped.

## 6. Recent Knowledge to be Taken into Consideration for Phage Therapy

The number of phage-related in vitro and in vivo studies, combined with newer areas of research such as the microbiome, has never been greater and provides a wealth of knowledge to keep in consideration when approaching clinical application. The activity of phages against biofilms, their ability to block bacterial receptors, and their synergy with conventional antibiotics has important implications for clinical treatment. Beyond bacterial lysis, phages have also been shown to interact in different ways with the immune system of the patient and their overall microbial community. The role that phages play naturally in the microbiome ecosystem is only starting to be discovered. Awareness and incorporation of these aspects provides a greater understanding of phage therapy and its clinical utility.

Regarding antibacterial aspects, the most pertinent aspect of new knowledge is the exploration of phage-antibiotic synergy (PAS). Several recent studies both in vitro [7,8,119,120] and in vivo on numerous experimental animal models [121] have confirmed the potential of combined use by showing the synergy of specific bacteriophage–antibiotic combinations at sometimes sub-inhibitory doses [122,123]. This could be a function of reducing the development of bacterial clones resistant to traditional antibiotics, by separate killing mechanisms, or other additive functions.

In almost all compassionate use cases, phages have been used in conjunction with antibiotics. It was even shown that phage administration changed the antibiotic resistance profile during the treatment of an *A. baumanii* infection, which led to the inclusion of the antibiotic in the treatment regimen [21]. Additionally, it has been shown [124] that, to combat *S. aureus* infections, the therapeutic results can also be influenced by the sequence in which the therapeutic agents are administered: best results were obtained when phage therapy precedes antibiotic therapy. As interesting as this effect may be, methods for determining the best choice of phage(s) and antibiotic(s) are still lacking. Nevertheless, the reintroduction of phage therapy deserves to be approached with the idea that it could be not only an alternative, but also a complement, in circumstances where the diffusion of an antibiotic is weak, as is the case in bone tissue or in the presence of a biofilm for example [6].

The activity of phages against bacterial biofilms is yet another factor in support of phage therapy. The pathogenic role of biofilms appears fundamental in chronic infections, especially in the presence of foreign materials (i.e., prosthesis, catheter). The proteolytic enzymes of certain bacteriophages are capable of destroying polysaccharides in biofilms which allow bacteria to escape natural defenses and antibiotic treatments [125]. In addition to allowing the adhesion of bacteriophages on the bacterial surface, this action facilitates the diffusion of antibiotics. It should be noted that soluble degradation products of *S. aureus* biofilm components could have a deleterious role on osteoblasts [126] and thus limit the growth of bone callus, which would explain the rapid bone healing observed after bacteriophage treatment observed in the compassionate cases in Table 1.

The very interaction of phages with the surface of bacterial cells may itself have an additive effect for phage therapy. It has been shown that bacteriophages, by attaching themselves to the bacterial surface at particular sites, could block resistance mechanisms such as an efflux pump or impair the fitness or the virulence factor of a bacterium [127]. This would then make certain bacteria (i.e., *P. aeruginosa* or *K. pneumoniae*) more susceptible to traditional antibiotics and facilitate the healing of certain pathologies, such as endocarditis or vascular prosthesis infections.

Regarding the interaction with mammalian cells, facets that are directly linked to the bactericidal effects are further complemented by a larger understanding of phage interaction with human cells and physiology, particularly with the immune system. Studies indicate that, in addition to their well-known antibacterial action, bacteriophages have potent immunomodulatory properties. For some authors, the success of phage therapy, depending on the bacterial permissiveness of the phage, is related to the immunity of the subject. In particular, for Roach et al. [46], neutrophil–bacteriophage synergy demonstrated that it is essential for the cure of pneumonia. For Dabrowska [128], the impact on the immune system affects the final outcome of phage therapy. While antibody induction may play a role in eliminating bacteriophages, it has also been shown that they can induce cytokine production in mammalian immune cells.

In reference to phages and surrounding microbiota, bacteriophages are present in all micro-ecosystems found in nature, and their presence in human microbiota is becoming increasingly recognized. Human microbiomes are distinct for various anatomical niches of the body (digestive tract, vaginal cavity, mouth, airway, nares, skin, urine [129,130,131,132,133,134,135]) that house dense microbial communities containing not only bacteria, archaea and fungi, but also mainly viruses, of which bacteriophages are the majority and remain largely unexplored [135,136,137]. The notion of the microbiome must be borne in mind when a bacteriophage treatment is being considered [138]. Indeed, the introduction of a bacteriophage in a structured community is not without consequence, because it induces difficult-to-predict interactions that may facilitate or hinder the intended effect [139]. Interactions occur not only with the microbiome with which it comes in contact, but also eukaryotic tissue cells [140,141] and the immune system of the host organism, as mentioned above [142]. Consequently, a model consisting of this "ménage à trois" has been proposed and should be considered [143]. A good knowledge of these components could help improve treatment outcomes and should make phage therapy a more personalized therapy.

An interesting consideration of gastro-intestinal diseases is the interplay with surrounding gut microbiota. Indeed, many gastro-intestinal diseases are increasingly described as a dysbiosis in the microbial community rather than being caused by a discrete pathogen, and microbiome sequencing has been useful in revealing disease-associated microbial signatures [144,145]. Phages may be useful in restoring a proper balance, such as for Crohn’s disease or ulcerative colitis, and a trial targeting Enteroaggregative *E. coli* (EAEC) has been initiated for Crohn’s patients [146,147,148].

Currently in Western countries, *Clostridium difficile* is a major problem (regarding diarrhea and transmission within the community), against which conventional antibiotics are ineffective. Many authors in recent years have considered addressing this condition with bacteriophages [149,150,151], and a study has shown a strong adsorption of bacteriophages on human cells in vitro that would promote bacteriophage–bacterial interactions is important for treating such a condition [152]. Fecal microbiota transplantations (FMT) have been shown to be effective at treating *C. difficile*, and, more so, filtrates of FMT that are devoid of bacteria also retain therapeutic properties, which may be due to the presence or modulation of phages [153].

## 7. Conclusions

Noting a continuing increase in bacterial resistance to antibiotics and the scarcity of new antibiotic molecules, the World Health Organization declared in 2014 that a pre-antibiotic era was imminent [154] and that there was an urgent need worldwide to mobilize international cooperation. In view of the risk to public health, new strategies need to be considered without delay: phage therapy is one of the most successful options today, if not the most successful. The advancement of phage therapy will, however, require an entwinement of old and new, of science and medicine, of fundamental research and clinical application, that is unparalleled in other areas of medical research. A multidisciplinary approach is needed more than ever, bringing together microbiologists, ecologists, evolutionary biologists, infectious disease specialists, medical doctors, and public health professionals. The growing threat of antibiotic resistance is indeed a compelling motivation to include and evaluate as much historical, compassionate use, and pre-clinical information as possible to increase the likelihood of the effective implementation of phage therapy.

The limited knowledge of phages available when they were first used historically has been complemented by a wealth of scientific studies, and yet therapy remains largely as empiric today as it was then. Our own empirical compassionate experiences with phages have nevertheless resulted in good clinical outcomes and have led us to conclude that phage therapy has much to offer, particularly for osteoarticular infections. A clinical trial is now planned to treat osteoarticular infections as an extension of our empirical findings through compassionate treatment. Such observational evidence from individual treatments provides valuable information on how to refine treatment protocols and to guide effective clinical practice in the future.

The use of biological rather than chemical drugs, such as phages, is new and upsets conventional treatment paradigms. Moreover, this development is occurring in a more strictly regulated context than in the past, where therapeutic frameworks need to be navigated and financial support is lacking. If it is unlikely that phage therapy will ever replace antibiotic therapy, it would surely be best to combine available antimicrobial strategies to create an effective treatment, and more research is merited in this direction. In the interim of conclusive phage efficacy trials, compassionate use of phage therapy, in combination with appropriate antibiotics, should be continued to maximize positive treatment outcomes for patients suffering from antibiotic resistant or difficult-to-treat infections.

## Figures and Tables

**Table 1 viruses-11-00018-t001:** Summary of patients treated with compassionate use of bacteriophages from 2006–2018.

N	Age; Sex	Symptom Onset; PT Start	Clinical Symptoms	Bacteria	Phage Therapy	Outcome
1	20; F	2004; 2006	Suppurating chronic otitis; intense pain	*S. aureus*	Commercial anti-*S. aureus* suspension; ear drop instillations (15 days)	2006 Complete cure
2	44; M	2005; 2008	Accidental fall; multiple fractures (*n* = 37); amputation considered	*S. aureus*	Commercial anti-*S. aureus* and Pyophage suspensions; administered peroperatively over several weeks	2009 Wound closure and complete cure
3	25; M	2007; 2008	Road accident causing multiple trauma; uncontrolled pelvic bone infection	*S. aureus* *P. aeruginosa*	Anti-*S. aureus* and anti-*P. aeruginosa* phage suspension; administered peroperatively and via catheter in days following operation (Belgium).	2010 Complete cure
4	40; F	1995; 2009	Fall leading to complex fracture of the right foot; Planned amputation	*S. aureus*	Commercial anti-*S. aureus* suspension administered peroperatively and via catheter in the days following operation	2009 Wound closure and complete cure
5	60; M	2008; 2009	Fistulised abdominal plaque infection; continuous suppressive antibiotic administration	Methicillin resistant *S. aureus* (MRSA)	Commercial anti-*S. aureus* suspension administered via fistula	2010 No recurrence without any antibiotic over 4 years
6	80; F	2008; 2010	Knee prosthesis infection unsuitable for surgery	*P. aeruginosa*	Commercial broad spectrum multi-bacteriophage suspension; Knee joint injection	2012 *P. aeruginosa* clearance, but appearance of *Enterococcus* sp.
7	61; F	1995/2005; 2010	Operated tongue cancer; Dental extraction, jaw fracture, osteo-synthesis and fistulised infection	*S. aureus* (MRSA)	Commercial anti-*S. aureus* suspension administered peroperatively	2011 Complete cure
8	90; F	2009/2010; 2010	Femoral fracture under hip prosthesis; Drained hematoma and antibiotherapy-infection	*S. aureus* (MRSA)	Commercial anti-*S. aureus* suspension administered peroperatively by flooding the infection site and via catheter in the 10 days following the operation	2011 Complete cure, rapid recovery without recurrence after 1 year with retention of the hip prosthesis and osteosynthesis material *in situ*
9	20; M	2012; 2012	Chronic Ulcerative Colitis with liver complications. Severe weight loss (54 kg down from 80 kg). Poor digestion of food.	*E. coli, Proteus* spp. *S. aureus* (Urine) *S. aureus* (skin) *E. coli, Proteus vulgaris, Proteus mirabilis* (stool)	Treatment in Tbilisi (Georgia) with 2 commercially available phage suspensions plus special customised phage suspension. Probiotics, enzymes and Camelyn immune stimulant also given. Treatment lasted 1 month.	2012 Healing with sterilisation of urine, reduction of *E. coli* and *P. vulgaris* growth from high (10^8^) to low (<10^2^) in stool. Weight gain to 72 kg by end of treatment. Digestion improved but still poor
10	72; F	2009; 2013	Left knee prosthesis infection	*Staphylococcus* sp.	Commercial anti-*S. aureus* suspension administered peroperatively by flooding the infection site	2013 Initial partial disinfection with closure of several fistula followed by stabilisation
11	84; M	1943/2012; 2013	Osteomyelitis of the left tibia; Fistula next to the wound	*S. aureus* (MRSA)	Initial phage therapy treatment in Tbilisi via fistula with temporary improvement, followed by surgical follow up intervention in France in 2013; Commercial anti-*S. aureus* suspension administered peroperatively by flooding the infection site	2013 Complete cure
12	58; F	2000; 2013	Acoustic neuroma with nosocomial infection of the ENT and ophthalmic regions	*S. aureus*	Treatment in Tbilisi with locally produced phage suspensions administered locally and orally	2013 Complete cure allowing an ophthalmic intervention of the retina that had been delayed for several years
13	68; F	1973; 2015	Operated left tibia fracture, followed by re-opened bone infection 2013: Travel to Phage Therapy Center (Tbilisi)	*S. aureus*	Surgery, phage therapy with commercial staphylococcal phage suspension, and antibiotherapy	2016 Disappearance of *S. aureus* replaced by *P. aeruginosa* & *Streptococcus constellatus*, followed by complete cure without recurrence
14	84; M	2006 & 2015; 2016	Prostate adenectomy with chronic urinary infection and bacteraemia	Extended-spectrum beta-lacatamase *E. coli* (ESBL)	Anti-*E. coli* phage suspension administered per os and rectally	2018 Complete cure
15	86; M	2016; 2018	Recurring prostatitis with bacteraemia	*P. aeruginosa*	Commercial multi-phage suspension administered orally and rectally	2018 Complete cure with disappearance of any urinary infection for the first time in 2 years

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
