# Peer review of "Clinical Indications and Compassionate Use of Phage Therapy: Personal Experience and Literature Review with a Focus on Osteoarticular Infections"

_viruses, 2018, doi:10.3390/v11010018_

Reviewer 1 Report

The authors appropriately present information directly relating to the development and usage of "phage therapy" for the treatment of difficult pathogenic bacteria. The authors broadly describe different aspects that impact the employment of phages, including regulatory issues, delivery modalities, and anticipated success expectancies. The review includes valuable information about currently operating companies, academic institutions, and clinical establishments devoted to the delivery of phage therapy. The article presents "phage therapy" in a positive light, and serves as as an excellent review of the current state of the field. 

There are no major points to be addressed

There are a few minor points to be addressed:

Line 155: “Amplify Biosciences” should be spelled “AmpliPhi Biosciences”

 Line 173: “… few cases have been published to date [41-46], ….” The author has failed to include one of the “few” original phage nebulization studies in this list:

Semler, D.D.; Goudie, A.D.; Finlay, W.H.; Dennis, J.J. Aerosol phage therapy efficacy in Burkholderia cepacia complex respiratory infections. Antimicrob. Agents Chemother. 2014, 58, 4005-4013. Instead, the author references review articles, one of which is in a now defunct journal (Abedon, ST). This is not wrong, but it is more acceptable to reference scientific studies rather than scientific reviews in a review.

 Line 216: Something is incorrect in Ref. 37. Remove the words “microbiology letters S.A. “.

Author Response

We greatly appreciate your consideration and critical review of our manuscript. Please find out responses  with reviewer comments written in blue and author responses, immediately following, written in black:

Line 155: “Amplify Biosciences” should be spelled “AmpliPhi Biosciences”. This was corrected.

Line 173: “… few cases have been published to date [41-46], ….” The author has failed to include one of the “few” original phage nebulization studies in this list:

Semler, D.D.; Goudie, A.D.; Finlay, W.H.; Dennis, J.J. Aerosol phage therapy efficacy in Burkholderia cepacia complex respiratory infections. Antimicrob. Agents Chemother. 2014, 58, 4005-4013. Instead, the author references review articles, one of which is in a now defunct journal (Abedon, ST). This is not wrong, but it is more acceptable to reference scientific studies rather than scientific reviews in a review. The reference was added [43-48].

Line 216: Something is incorrect in Ref. 37è39. Remove the words “microbiology letters S.A. “. The words were removed

Reviewer 2 Report

This manuscript by Dublanchet et al is a comprehensive, well-referenced review of bacteriophage therapy, adding the authors' experience to  this field.   The review highlights the difficulties in this realm - the lack of rigorous studies, well-controlled clinical trials and a difficult regulatory/commercial landscape.

The new data presented is interesting as a  15 patient case series, supporting the role of phage therapy in Staphylococcal bone/joint infections (n=9)  and possibly prostatitis (n=2), Staph aureus ear infections (n=2) and abdominal wounds (n=1). Its not clear why case #9 was included -it was not treated by the authors, and the description of the infectious site/process/organism ("ulcerative colitis with live complications") and treatments rendered was confusing at best. 

It appears treatment of the targeted pathogen was effective in all cases, but 2 of the bone/joint infections had clinical relapse with different pathogens.

While the authors emphasize the additive/synergistic role of antibiotics with phage therapy, their biggest omission is not including concurrent antibiotic treatment/duration in their data table. Complex regimens could be included in footnotes, but the duration of cumulative antibiotic therapy after phage therapy was started would be most helpful in comparing the results to "standard" therapies for these types of patients, in the absence of a comparator group. Similarly, the number of prior antibiotic "courses"  could be cited, to provide a type of internal/historical control for the efficacy of the phage therapy approach.

While there were no clear failures of therapy here, it would be informative to know whether failures that did occur were due to phage-resistant variants, confirming a biological selection process.

--------------------------------------------------------------------

The text was generally well written.

Some comments for clarification:

Page 7, line 270 - "Treatment failure with added phages caused a change in treatment plan, as needed." This is unclear - was the change in treatment involving new phage preparations or conventional antibiotics.  Were the 2 failures of treated bone infections due to phage-resistant variants?

Page 7, line 290 - "evolutionary" should be "evolving"

Page 9, line 357-358 - "suppression of the infectious process during gram-negative pediatric epidemics" -Could this be explained more informatively - what sites of infection (bacteremias, UTI's) and which organisms

Page 9, line 375-377 -." During a discharge consultation with acute pain and under general anaesthesia, a specimen was collected showing the presence in pure culture of S. aureus (penicillin-R, methicillin-R, erythromycin-R and ofloxacin-R). Despite antibiotic therapy immediately prescribed," - Replace "discharge" with "outpatient".  Identify the "antibiotic therapy"

Table -case 9 - "Ulcerative colitis/chronis" - I am unfamiliar with the term "Chronis" - is this what was meant or is it chronic

Author Response

We greatly appreciate your consideration and critical review of our manuscript. Please find out responses with reviewer comments written in blue and author responses, immediately following, written in black: 

From the general comments provided by Reviewer 2, it would have been desired to include antibiotic data. We agree entirely that more information on the antibiotic regimen could be useful information to include in the manuscript. Unfortunately, the medical records were not readily available at this time and would cause a significant delay. Additionally, there was no microbiological data available on phage resistant variants for the past paper or compassionate cases. The authors agree that this should be reported for compassionate cases, but it was not systematically done for the presented cases here.

Its not clear why case #9 was included -it was not treated by the authors, and the description of the infectious site/process/organism ("ulcerative colitis with live complications") and treatments rendered was confusing at best. This case was included because it was supported and followed by one of the authors; also it signifies a wider range of phage therapy indications.

It appears treatment of the targeted pathogen was effective in all cases, but 2 of the bone/joint infections had clinical relapse with different pathogens. A summary sentence was added: “In fact, 12/15 cases resulted in a complete recovery (a secondary problematic pathogen emerged in one case, only a stable condition was achieved for one patient, and one case of a GI infection improved after phage treatment, but for which the condition was not fully resolved).”

Page 7, line 270 - "Treatment failure with added phages caused a change in treatment plan, as needed." This is unclear - was the change in treatment involving new phage preparations or conventional antibiotics.  Were the 2 failures of treated bone infections due to phage-resistant variants? The specific treatment plan for the one phage failure was specified. Unfortunately, there is no data on phage-resistance of the clinical isolates.

Page 7, line 290 - "evolutionary" should be "evolving". This was corrected.

Page 9, line 357 - "suppression of the infectious process during gram-negative pediatric epidemics" -Could this be explained more informatively - what sites of infection (bacteremias, UTI's) and which organisms This indicated pathogens were added. The authors do not specify the sites of infection, but indicate that oral phage application was used to reduce the infectious process. These manuscripts are currently available only in French (several authors are bilingual), but we intend to have the professionally translated and published in the near future.

Page 9, line 375 -." During a discharge consultation with acute pain and under general anaesthesia, a specimen was collected showing the presence in pure culture of S. aureus (penicillin-R, methicillin-R, erythromycin-R and ofloxacin-R). Despite antibiotic therapyimmediately prescribed," - Replace "discharge" with "outpatient".  Identify the "antibiotic therapy". The wording was corrected; information concerning the specific antibiotic therapy however was not specified as medical records were not available at this time.

Table -case 9 - "Ulcerative colitis/chronis" - I am unfamiliar with the term "Chronis" - is this what was meant or is it chronic. This was clarified in the text.

Reviewer 3 Report

This article "Clinical indications and compassionate use of phage therapy: Personal experience and literature revieuw with a focus on osteoarticular infections" is surely an interesting paper especially for clinicians wanting to use phages as a sensefull antibacterial treatment. Especially in times of increasing antibiotic resistances. What makes this paper very interesting is , in my opinion, what the authors mention in their abstract: "...It would then indeed be wise to identify why such a discordance exists between trials and compassionate use in order to better shape (I would say develope) future phage treatment and clinical development (I would say applications)" (line 20-21).

Being not a native English speaking person I have some phrasing issues. But seen that some of the authors are native English speakers I will not comment so much this issue. But for example the last sentence of the abstract  (line 28) I would reformulate it into: ...seems conceivable and scientifically sensefull today as a substitute or adjuvant to antibiotic therapy. 

A practical remark that confused me is the fact that the numbering of lines and the page numbers of the document are not consistent. So, sorry if some of my remarks are not always easy to find back based on the line numbers. For example the title  "6. recent knowledge to be taken into consideration for phage therapy" has no page number and the lines restart at 0 till 511 , the last reference.

P 2 line 43: Please delete the word 'common" as well as "several" since so far as I personally experienced and asked the Polish colleagues during the last Viruses of Microbes congress "phage therapy" is also in Poland "not so common!" as we would think. Therefore I suggest to adapt the sentence into:

However, phage therapy was continued uninterupted in the Soviet Union during this time and is actually still practiced in some East European or former Soviet States today, although in accordance of  specific national regulations.

A question: in line 41 it is said that phages in France were in use or available untill the late 1980's while in the book of dr Alain Dublanchet ( Des virus pour combattre les infections) he says untill early 1990. Please justify or make it more correct.

Line 49, please add to ref 6 also the refs. 115 and 116. This because the biologically evolutionary rationale of the combination of both has to be emphasized.

Plaese as a general rematk check that all (bacterial) species names are put in italics all through the text. Sometimes it is sometimes not: so please Escherichia coli  in line 51.

Important remark on line 59 -60: "Phages ( understand: "natural" and not engineered phages ) are currently classified as medicinal products under actual European legislation (ref 12!) and as a drug in the US. Thus delete the ATMP classification although some people are trying or lobbying to put phages under the ATMP classification. ATMP's could become PTMPs as Phage Medicinal Therapeutic Products.

Box 1:

Please rephrase and check content of lines 86 and further:

So far as I understood after discussions with the Polish colleagues " Phage therapy is still practiced in Poland, a EU member state, but under strict regulation. This means it was seen as an established treatment when Poland became member of the EU and was allowed to continue this type of treatment although under a National regulatory frame "as experimental therapy in accordance to some medical issues like resistant bacteria and in specific phage centre. In fact out of the Article 37 of the Helsinki Declaration. Please check specifically in order to give the correct information.

Line 93:

Magistral preparations in Belgium (also known as compounded prescription drug in the US) (ref 12).

Line 96: Please add abreviation after: " Phages are considered active pharmaceutical ingredients (APIs) ...

Line 102: Acinetobacter baumannii 

Line 104: ... to be treated Intravenously (IV) by phage therapy and successfully. 

Line 114: please delete "largely" since it is not so common. 

Line 118: ...the patient's isolate...Line 125: I would delete  "available for purchase"  Maybe a more or better rephrasing of the paragraph would be better...

Line 141:

...products, they are are subject to classic GMP compliance ... I would suggest to rephrase the sentence with emphasis on the fact thet the actual GMP procedures, developed essentially during the classic and static drug (antibiotic) development is not suited for phages which are (co)evolving biological entities . So a flexible and adapted GMP frame has to be developed for phage therapy in order to take into account the sustainability of this treatment approach and the patients safety.

Line 145: strain? is not it meant "a constrain" a "barrier"?

In fact the Phagoburn study showed clearly that the actual system could not deliver good active phage products under the actual GMP procedure. Thus adaptations are urgently required.

Line 152: ...phage preparations are individually prepared on a MD prescription for a nominated individual patient and prepared by a pharmacist , actually stil only in a hospital pharmacy...

Line 179: Please add some reference of Steve  Abedon on the pharamcokinetics of phages: 

Line 219: Is not it meant "mucoid" instead of mucosal?231: please add reference

Line 259 please S. aureus in italics

Line 328: please specify 'which infections'

Under paragraph  6. Recent knowledge to be taken into consideration for phage therapy:

Please check the references on their correctness:

all refs mentioning "Vos, D" should be corrected into "De Vos, D"

Ref119: please complete or correct Oechslin, F.; etal: ..., of..., M.S. Oechslin et al

Being not an native English speaking person I have not "corrected" in that sense, but some things are not sounding well in my opinion, therefore please ask to revise it in that sense. Some of the co-authors are native English speakers.

Author Response

We greatly appreciate your consideration and critical review of our manuscript. Please find out responses, with reviewer comments written in blue and author responses, immediately following, written in black:

It would then indeed be wise to identify why such a discordance exists between trials and compassionate use in order to better shape (I would say develope) future phage treatment and clinical development (I would say applications)" (line 20-21è26). Shape was changed to develop and clinical development was changed to clinical application.

But for example the last sentence of the abstract  (line 28è33) I would reformulate it into: ...seems conceivable and scientifically sensefull today as a substitute or adjuvant to antibiotic therapy. Scientifically sound was used instead of merited.

A practical remark that confused me is the fact that the numbering of lines and the page numbers of the document are not consistent. So, sorry if some of my remarks are not always easy to find back based on the line numbers. For example the title  "6. recent knowledge to be taken into consideration for phage therapy" has no page number and the lines restart at 0 till 511 , the last reference. We apologize for this inconvenience and believe it to have arisen from an error in formatting.

P 2 line 43: Please delete the word 'common" as well as "several" since so far as I personally experienced and asked the Polish colleagues during the last Viruses of Microbes congress "phage therapy" is also in Poland "not so common!" as we would think. Therefore I suggest to adapt the sentence into: However, phage therapy was continued uninterupted in the Soviet Union during this time and is actually still practiced in some East European or former Soviet States today, although in accordance of  specific national regulations. This was specification was added, with minor modifications for English.

A question: in line 41 it is said that phages in France were in use or available untill the late 1980's while in the book of dr Alain Dublanchet ( Des virus pour combattre les infections) he says untill early 1990. Please justify or make it more correct. This was changed to "1990".

Line 49, please add to ref 6 also the refs. 115 and 116. This because the biologically evolutionary rationale of the combination of both has to be emphasized. They were added.

Plaese as a general rematk check that all (bacterial) species names are put in italics all through the text. Sometimes it is sometimes not: so please Escherichia coli  in line 51. Bacteria names, as well as in vitro/in vivo, were italicized throughout the text.

Important remark on line 59 -60: "Phages ( understand: "natural" and not engineered phages ) are currently classified as medicinal products under actual European legislation (ref 12!) and as a drug in the US. Thus delete the ATMP classification although some people are trying or lobbying to put phages under the ATMP classification. ATMP's could become PTMPs as Phage Medicinal Therapeutic Products. ATMP was changed to medicinal product.

So far as I understood after discussions with the Polish colleagues " Phage therapy is still practiced in Poland, a EU member state, but under strict regulation. This means it was seen as an established treatment when Poland became member of the EU and was allowed to continue this type of treatment although under a National regulatory frame "as experimental therapy in accordance to some medical issues like resistant bacteria and in specific phage centre. In fact out of the Article 37 of the Helsinki Declaration. Please check specifically in order to give the correct information. This information was added.

Line 93: Magistral preparations in Belgium (also known as compounded prescription drug in the US) (ref 12). This was changed.

Line 96: Please add abreviation after: " Phages are considered active pharmaceutical ingredients (APIs). The abbreviation was added.

Line 102Acinetobacter baumannii. This was italicized.

Line 104: ... to be treated Intravenously (IV) by phage therapy and successfully. The specification of IV was added.

Line 114: please delete "largely" since it is not so common. Largely was changed to mostly – it is not to emphasize the frequency, but the geographical concentration of where compassionate treatment happens.

Line 118: ...the patient's isolate...Line 125: I would delete  "available for purchase"  Maybe a more or better rephrasing of the paragraph would be better... This comment is not clear – “patient’s bacterial isolate” was specified and “available for purchase” was maintained as this is how the products were obtained for the following case reports (as well as for the E. coli diarrhea trial and for metagenomic analysis). We apologize to the reviewer if we misunderstood this comment.

Line 141: ...products, they are are subject to classic GMP compliance ... I would suggest to rephrase the sentence with emphasis on the fact thet the actual GMP procedures, developed essentially during the classic and static drug (antibiotic) development is not suited for phages which are (co)evolving biological entities . So a flexible and adapted GMP frame has to be developed for phage therapy in order to take into account the sustainability of this treatment approach and the patients safety. This is a valid point, but the authors do not wish to enter into discussion about production regulation, as this is not our field of expertise. However, a sentence was added “Production considerations must take into account the sustainability of the phage treatment approach and patient safety,” in order to emphasize the concern of the reviewer.

Line 145: strain? is not it meant "a constrain" a "barrier"? Strain in this sense means “a force causing damage” or “a severe demand on resources”. Strain was maintained in the text.

 In fact the Phagoburn study showed clearly that the actual system could not deliver good active phage products under the actual GMP procedure. Thus adaptations are urgently required. This point was added “Indeed, GMP constraints both delayed patient enrollment for the Phagoburn clinical study, and negatively impacted the phage titer of the final product.”

Line 152: ...phage preparations are individually prepared on a MD prescription for a nominated individual patient and prepared by a pharmacist , actually stil only in a hospital pharmacy... Hospital was specified.

Line 179: Please add some reference of Steve  Abedon on the pharamcokinetics of phages: An appropriate reference was Added ].

Line 219: Is not it meant "mucoid" instead of mucosal?231: please add reference. Mucosal is referring to the anatomical origin or these bacteria in this instance. Mucosal was maintained in the text and a reference added.

Line 259 please S. aureus in italics. This was italicized.

Line 328: please specify 'which infections' Infections were specified (2 ENT, 1 abdominal, 1 GI).

Please check the references on their correctness: all refs mentioning "Vos, D" should be corrected into "De Vos, D" Ref119: please complete or correct Oechslin, F.; et al: ..., of..., M.S. Oechslin et al The references were corrected.